# CCL: Causal-aware In-context Learning for Out-of-Distribution Generalization

**Hoyoon Byun, Gyeongdeok Seo, Joonseong Kang, Taero Kim, Jihee Kim, Kyungwoo Song**[*]
Department of Statistics and Data Science, Yonsei University
{hoyun.byun, gd.seo, doongsae, taero.kim, jihee_sta, kyungwoo.song}@yonsei.ac.kr

## Abstract

In-context learning (ICL), a nonparametric learning method based on the knowledge of demonstration sets, has become a de facto standard for large language models (LLMs). The primary goal of ICL is to select valuable demonstration sets to enhance the performance of LLMs. Traditional ICL methods choose demonstration sets that share similar features with a given query. However, our experiments reveal that these traditional ICL approaches perform poorly on out-of-distribution (OOD) datasets, where the demonstration set and the query originate from different distributions. To ensure robust performance in OOD datasets, it is essential to learn causal representations that remain invariant between the source and target datasets. Inspired by causal representation learning, we propose causal-aware in-context learning (CCL). CCL captures the causal representations of a given dataset and selects demonstration sets that share similar causal features with the query. To achieve this, CCL employs a novel VAE-based causal representation learning technique. We demonstrate that CCL improves the OOD generalization performance of LLMs both theoretically and empirically. Code is available at: https://github.com/MLAI-Yonsei/causal-context-learning

## 1 Introduction

While large language models (LLMs) excel as general-purpose pre-trained models, in-context learning (ICL) has become a key approach for aligning them to target tasks. ICL [1] enables LLMs to adapt to new tasks with a few demonstrations and without parameter updates, making it applicable in various fields. While ICL has shown significant promise, it still faces difficulties in achieving robust generalization [2]. A primary challenge is that LLMs rely on superficial patterns in demonstration sets, which restrict their capability in unseen environments [3]. Recent studies indicate that distribution shifts between demonstration sets and target queries in out-of-distribution (OOD) scenarios impede the ability of LLMs to generalize effectively [4, 5, 6]. To fully unlock the potential of LLMs and enable reliable deployment in real-world applications, ensuring robustness in OOD scenarios plays a pivotal role.

The pursuit of ensuring generalization beyond observed data naturally leads to the question of how the data was generated. Drawing on insights from causality [7], the structural knowledge of data is expressed using causal language. In causal representation learning (CRL) [8], observed data reflect underlying latent causal variables that drive the data-generating process (DGP). CRL aims to model the causal mechanisms among these variables [9]. For example, if two causal variables are independent, one remains invariant even when the other, acting as an environmental factor, changes [10, 11]. The assumption about causal mechanisms suggests that learning invariant causal variables is an effective approach for models robust to distribution shifts. Consequently, CRL lays the groundwork

---

[*]Corresponding author

39th Conference on Neural Information Processing Systems (NeurIPS 2025).

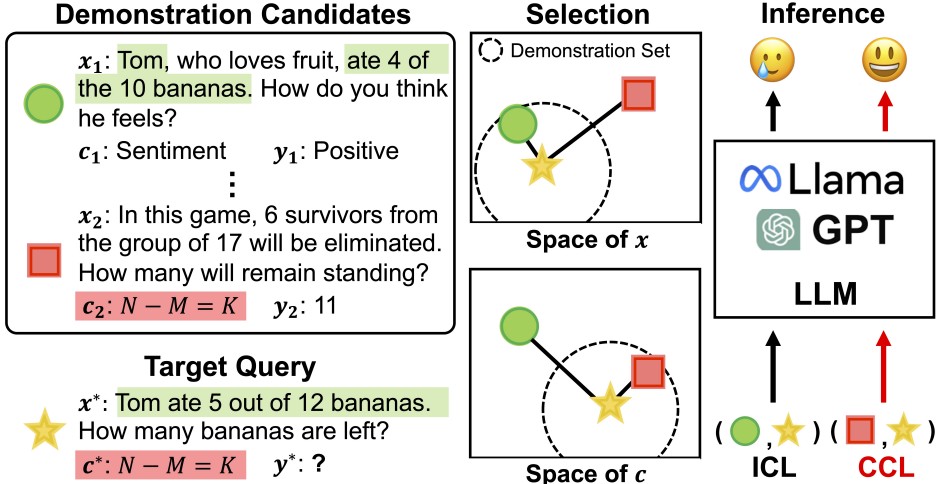

Figure 1: To enhance the OOD performance of LLM, a causally-related demonstration set is important. Current ICL methods compare the non-causal representation $x_i$ and $x^*$, and they might choose a worthless demonstration set (Candidate 1). However, our method, CCL, compares the causal representation $c_i$ and $c^*$ to construct a demonstration. Because CCL leverages the causal-related demonstration (Candidate 2), CCL shows superior performance on the OOD dataset.

for research on OOD generalization by learning invariant representations from training sets collected across multiple environments. [12, 13, 14].

Considering causal mechanisms allows for constructing a more suitable demonstration set from demonstration candidates when their environments differ from that of the target query. In Figure 1, the target contextual problem $x^*$ is highly similar to $x_1$ (highlighted in green), making $x_1$ a strong candidate in ICL [15, 16]. But what exactly is the *problem* embedded in the target query? For LLMs to successfully generalize to the target query, the demonstration set should be constructed to reflect the fundamental context rather than relying on superficial patterns, such as frequently occurring words or characteristics of the data collection environment [17].

Therefore, even if the superficial context differs, a candidate $x_2$ that addresses the same problem (*N-M=K*) should be included in the demonstration set (highlighted in red). It ensures the demonstration set captures problem-level invariance even when generalizing to OOD targets from given candidates. Since causal variables $c$, which generate the contextual problem, are not observable objects, it is necessary to model $c$ under the assumption of causal mechanisms that remain invariant across environments.

In this study, we focus on constructing a robust demonstration set to enhance the generalization of LLMs in OOD scenarios. Inspired by CRL, we propose a novel demonstration selection method, causal-aware in-context learning (CCL), which learns causal representations that remain invariant across environments and prioritizes candidates by assigning higher ranks to those with causal representations similar to the target query. Under the causal mechanism, we theoretically demonstrate that the demonstration set selected by CCL comprises candidates that are more closely related to the underlying problem addressed by the target query, rather than merely matching its context. The problem-level invariance of CCL ensures generalization performance for the target query even in unseen environments. We empirically validate that CCL operates robustly in OOD scenarios and demonstrates superior generalization performance on both synthetic and real datasets.

## 2 Related Works

### 2.1 In-context learning

ICL is a method where LLMs perform tasks by leveraging examples from the input context without updating model parameters [1]. This approach enhances computational efficiency and achieves competitive performance in various natural language tasks without the need for model fine-tuning

[2, 18]. However, the performance of ICL is sensitive to demonstration organization, including demonstration selection [19, 20]. Various approaches aim to optimize demonstration selection in ICL, including unsupervised methods that use similarity metrics like k-nearest neighbors [15], as well as supervised techniques that leverage task-specific retrievers [21] and reinforcement learning [22].

Despite these advancements, LLMs depend on surface-level patterns in the demonstration set, leading to a primary challenge with out-of-distribution (OOD) examples [3]. While larger models tend to reduce the performance gap between in-distribution (ID) and OOD scenarios, even transformers, which handle minor distribution shifts, face significant challenges when encountering major shifts [6, 4]. The BOSS benchmark evaluates OOD robustness in ICL, highlighting the importance of addressing OOD generalization [5]. An approach designed to improve OOD performance involves inferring latent variables from the context using the transformer architecture. However, this method struggles to apply those variables effectively in prediction, limiting OOD generalization [23]. We propose CCL, drawing on causal representation learning, to improve OOD performance in ICL by focusing on task-relevant causal features and enhancing robustness to distribution shifts.

## 2.2 Causal representation learning

Unlike statistical approaches, which describe the distributional characteristics of data, causality [7] focuses on the structural relationships between variables. The DGP is determined by the underlying causal relationships among variables, and a structural causal model (SCM) is a generative model that describes the DGP [10, 24]. The SCM expresses the uncertainty of exogenous factors in a probabilistic manner and defines functional relationships for the variables of interest (endogenous variables), thus structurally describing the causal mechanisms of the DGP. Observed data represent one of the realizations of these causal mechanisms. A causal graph visually represents the structural relationships between the variables, as induced by the SCM [7].

Recently, research in machine learning has increasingly focused on moving beyond models limited to statistical associations [25], aiming to model the underlying structural properties of the data by applying the causality framework to machine learning [26, 27]. CRL aims to construct latent variables that capture the underlying causal mechanisms, allowing for the discovery of causal representations within observed data [8]. It seeks to deploy robust models in OOD scenarios, ensuring reliable performance even when the data distribution shifts. For example, leveraging the stability of causal mechanisms across different environments, several studies have utilized the invariant properties of causal representations under distribution shifts to enhance model performance in OOD scenarios through invariant prediction [12, 28].

Furthermore, there has been ongoing research into utilizing deep generative models to explicitly represent causal variables. Notably, under the assumption of independent causal mechanisms [10], several studies have modeled these mechanisms as separate, independent modules or have focused on learning disentangled and interpretable representations [11, 29, 30]. Research has evolved toward learning causal representations that maintain stable mechanisms under distribution shifts, to improve OOD generalization [13]. Inspired by CRL, we construct a novel ICL framework using causal knowledge for OOD generalization. To build a robust demonstration set, we utilize the invariant causal representation constructed by a Variational Autoencoder (VAE) [31]–based model [13, 32].

## 3 Methodology

### 3.1 Generative model and inference model

In CCL, we consider several key variables: the task variable $t$ represents the specific task being performed. The latent causal variable $c$ represents the fundamental context of the query. It is generated from the task variable $t$ and serves as a causal factor for both the input query $x$ and the (ground truth) answer $y$. Additionally, we introduce the latent source variable $s$, which influences components of $x$ that are unrelated to the task, such as the structure of the text. The environmental variable $e$ acts as an observable proxy for the latent source variable $s$. It represents contextual attributes of the data, such as the dataset's origin or the language used.

Note that both latent variables, $c$ and $s$, generate $x$, where $c$ represents task-specific information, and $s$ represents domain-specific information. That is, we assume that the domain shift in the observed data is induced by changes in $s$, while $c$ remains invariant, as shown in Figure 2.

Phase 1: Causal representation learning with In-pool (In-distribution) dataset

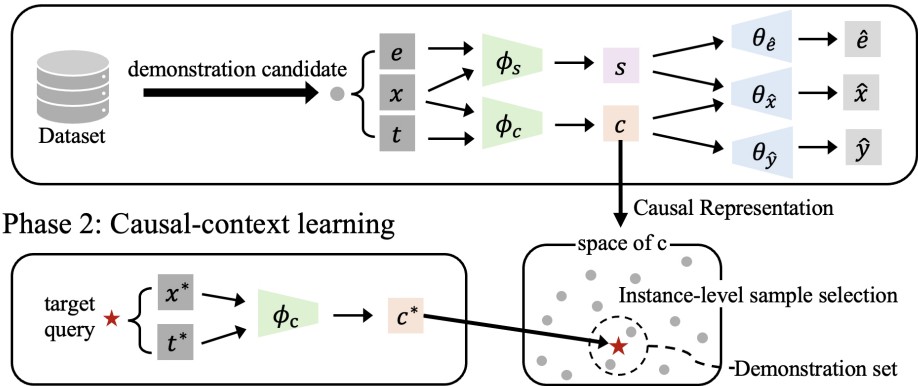

Phase 2: Causal-context learning

Figure 3: Our proposed method, Causal-aware In-Context Learning (CCL), utilizes causally related demonstration sets to enhance performance on out-of-distribution (OOD) datasets. (Phase 1) First, we optimize a novel VAE-based causal representation learning method to capture the causal representations of a given in-distribution dataset. After optimization, we store the causal representations, $c$, produced by the optimized model for the in-distribution dataset. (Phase 2) Second, CCL captures the causal representation, $c^*$, of the target query and selects the appropriate demonstration sets by comparing $c$ and $c^*$.

We aim to model the joint distribution of observed variables $\{x, y, t, e\}$ along with latent variables $\{c, s\}$. We assume the generative model

$$p_\theta(x, y, t, e, c, s) = p_\theta(x, t, e, c, s)\, p_\theta(y \mid c),$$

where $p_\theta(y \mid c)$ is an invariant causal mechanism. We let $\theta$ denote all parameters of the generative model. We denote the unknown true source-domain distribution as $p_{\theta^*}(x, y, t, e)$, and we approximate it with $p_\theta(x, y, t, e)$.

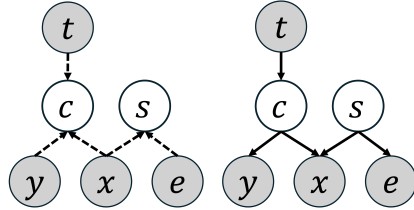

(a) Inference model  (b) Generative model

Figure 2: Graphical model of CCL. The generative model shows that $t$ influences the latent causal variable $c$, which in turn directly affects both $x$ and $y$.

Figure 3 illustrates the overall workflow of CCL in two phases. In Phase 1, we learn causal representations from an in-distribution (ID) dataset using our VAE-based model: the inference networks $\phi_s$ and $\phi_c$ infer the latent variables $s$ (environment-related) and $c$ (task-related), respectively, while the decoders $\theta_{\hat{e}}, \theta_{\hat{x}}, \theta_{\hat{y}}$ reconstruct the observed variables. This process yields the causal embeddings $c$ for the ID data. In Phase 2, given a target query $(x^*, t^*)$, we apply $\phi_c$ to obtain its causal embedding $c^*$. Comparing $c^*$ with the stored causal embeddings $c$, CCL then selects the most relevant demonstration examples, those with similar causal factors, to construct the prompt context. This causal representation approach ensures that our examples align with the true causal structure of the query, thereby improving model performance even under distribution shifts.

### 3.2 Learning causal representations via variational inference

Since direct maximization of $\log p_\theta(x, y, t, e)$ is often intractable due to the latent variables, we employ variational inference. We introduce a tractable inference model $q_\phi(c, s \mid x, y, t, e)$, where $\phi$ are the variational parameters. The standard Evidence Lower BOund (ELBO) on $\log p_\theta(x, y, t, e)$ is:

$$\log p_\theta(x, y, t, e) = \log \int p_\theta(x, y, t, e, c, s)\, dc\, ds = \log \mathbb{E}_{q_\phi(c, s \mid x, y, t, e)}\left[\frac{p_\theta(x, y, t, e, c, s)}{q_\phi(c, s \mid x, y, t, e)}\right]$$

$$\geq \mathbb{E}_{q_\phi(c, s \mid x, y, t, e)}\left[\log \frac{p_\theta(x, y, t, e, c, s)}{q_\phi(c, s \mid x, y, t, e)}\right] := L_{\text{ELBO}}$$

Maximizing this ELBO with respect to both $\theta$ and $\phi$ yields a tight approximation when $q_\phi(c, s \mid x, y, t, e) \approx p_\theta(c, s \mid x, y, t, e)$.

Since $\theta^*$ is unknown, we instead optimize the ELBO using the observed data distribution in the source domain, $p_D(x, y, t, e)$:

$$\max_{\theta, \phi} \mathbb{E}_{(x,y,t,e) \sim p_D(x,y,t,e)} [L_{\text{ELBO}}] \tag{1}$$

### 3.2.1  Reformulating variational inference for unobserved $y$

At test time, $y$ is always unobserved, as it is the target variable we aim to infer. While one common approach, such as in CEVAE [33], is to introduce an auxiliary model to explicitly predict $y$, we instead modify the objective function to enable variational inference without conditioning on $y$. Specifically, we factorize the inference model:

$$q_\phi(c, s, y \mid x, t, e) = q_\phi(c, s \mid x, t, e) \, p_\theta(y \mid c),$$

which reflects the conditional independence $y \perp (x, t, e, s) \mid c$. This design is key, as it directly injects the generative model's causal assumption ($c \rightarrow y$) into the inference process. It serves to constrain the inference model $q_\phi$ to find a $c$ that is consistent with $p_\theta(y \mid c)$, the actual causal mechanism from the generator. This formulation allows us to marginalize out $y$. By applying this factorization and Bayes' rule to the standard ELBO, we analytically marginalize out the unobserved $y$, reformulating the objective to depend only on $q_\phi(c, s \mid x, t, e)$ (see Appendix A for the full derivation). We define $\Phi_{y|x,t,e} = \mathbb{E}_{q_\phi(c,s|x,t,e)}[p_\theta(y|c)]$ as the implicit predictive distribution of $y$. The final objective of CCL is given by:

$$\max_{\theta, \phi} \mathbb{E}_{p_D(x,y,t,e)}[L_{\text{ELBO}}] = \mathbb{E}_{p_D(x,y,t,e)} \Big[ \log \Phi_{y|x,t,e} \\ + \frac{1}{\Phi_{y|x,t,e}} \mathbb{E}_{q_\phi(c,s|x,t,e)}[p_\theta(y|c) \times \log \frac{p_\theta(x,t,e,c,s)}{q_\phi(c,s|x,t,e)}] \Big]. \tag{2}$$

We construct the reconstruction model $p_\theta$ following the generative structure outlined in Figure 2b. Implementing Equation 2 requires this model, which is composed of decoders (e.g., $p_\theta(x \mid c, s)$, $p_\theta(y \mid c)$, $p_\theta(e \mid s)$) that reconstruct the observed variables from the latent variables. This reconstruction process, particularly the $p_\theta(y \mid c)$ mechanism, ensures that the learned causal representation $c$ effectively captures task-relevant information.

### 3.3  Regularization and conditional prior

In practice, to prevent unintended dependencies between $c$ and $s$ during training, we further employ Maximum Mean Discrepancy (MMD) [34] loss as a regularization term [9]. Additionally, the task variable $t$ (the parent of $c$) is treated as an observed input, not a latent variable requiring posterior inference. Instead, following the iVAE [35] framework, we define a conditional prior $p_\theta(c \mid t)$ for the generative model based on this observed $t$. Our variational inference formulation follows the approach proposed in [32].

### 3.4  Theoretical analysis

Prioritizing demonstrations that are causally similar to the query yields provably better in-context learning (ICL) than prioritizing demonstrations that are merely input similar. We show that input nearest selection can induce large label discrepancies even when inputs are arbitrarily close in Theorem 3.3. Furthermore, Theorem 3.4 provides both a theoretical explanation and a practical guideline: prioritizing causally similar examples is key to robust ICL.

Our analysis begins by assuming the data-generating process is modeled using an SCM $\mathcal{M} := (\mathcal{S}, P_\varepsilon)$ and a collection $\mathcal{S}$ of assignment equations as follows [10]:

$$t := \varepsilon_t, \quad c := f_c(t, \varepsilon_c), \quad s := \varepsilon_s, \quad e := f_e(s, \varepsilon_e), \quad x := f_x(c, s, \varepsilon_x), \quad y := f_y(c, \varepsilon_y). \tag{3}$$

Here, $\varepsilon_t, \varepsilon_c, \varepsilon_s, \varepsilon_e, \varepsilon_x \in \mathbb{R}^d$ are random vectors with $d \geq 2$ and $\varepsilon_y \in \mathbb{R}$ is a random variable. We assume $\varepsilon = \{\varepsilon_t, \varepsilon_c, \varepsilon_s, \varepsilon_e, \varepsilon_x, \varepsilon_y\}$ satisfies joint independence. The parents of $x$ are $c$ and $s$, while $y$ has only $c$ as its parent. The causal graph is achieved by drawing edges from RHS variables of Equation (3) to LHS variables except the noise variables $\varepsilon$.

We adopt a linear setting in line with [36], who demonstrate that attention-based updates in LLMs can be approximated by steps of gradient descent with a convex loss on a linear parameter $w$ with respect to $w^\top x$. Although real-world LLMs are more complex, the linear approximation provides a clear analytical framework.

**Assumption 3.1** (Linear-causal assumption). We formalize a simplified data-generating process via the following linear-causal assumption:

$$x_i := \mathcal{B}_1\, c_i + \mathcal{B}_2\, s_i + \varepsilon_{x,i}, \quad y_i := (w^*)^\top c_i + \varepsilon_{y,i}.$$

Each coordinate of $\varepsilon_{x,i}$ is $\sigma_x^2$-sub-Gaussian, and $\varepsilon_{y,i}$ is $\sigma_y^2$-sub-Gaussian. $\mathcal{B}_1$ and $\mathcal{B}_2$ denote coefficient matrices to $c_i$ and $s_i$. $x_i, c_i$, and $s_i$ are $d$-dimensional vectors and $y_i$ is a scalar.

A prerequisite for ICL is to construct a demonstration set $\mathcal{D}_S = \{(x_i, y_i)\}_{i \in S}$ from the training dataset $\mathcal{D}_T = \{(x_i, y_i)\}_{i \in \mathcal{I}}$, where $S \subset \mathcal{I}$ is the selected index set. A common strategy, forming the set $\mathcal{D}_x$, selects pairs $(x_i, y_i)$ by assessing how similar $x_i$ is to the input query $x^*$, with the expectation that $y^*$ will be similar to $y_i$ [15]. Our method leverages latent causal variables: we associate the query $x^*$ with a causal variable $c^*$, and each training pair $(x_i, y_i)$ with its own causal variable $c_i$. We then select pairs whose $c_i$ lie close to $c^*$, forming a causally similar set $\mathcal{D}_c$. All our theorems are based on Assumption 3.1.

**Definition 3.2** (Demonstration sets by strategy). Let $sim(\cdot, \cdot)$ be a similarity measure and $N$ the size of the demonstration set. We define:

$$\mathcal{D}_c = \underset{S \subset \mathcal{I}, |S| = N}{\operatorname{argmax}} \sum_{i \in S} sim(c_i, c^*), \quad \mathcal{D}_x = \underset{S \subset \mathcal{I}, |S| = N}{\operatorname{argmax}} \sum_{i \in S} sim(x_i, x^*) \tag{4}$$

**Theorem 3.3** (Input proximity can lead to prediction discrepancy). *Let $(x^*, y^*)$ and $(x, y)$ be two samples potentially generated by different latent pairs $(c^*, s^*)$ and $(c, s)$. Under Assumption B.1, B.2 in Appendix B, for every $\epsilon > 0$, there exists a $\kappa > 0$ such that*

$$\|c^* - c\| > \frac{\kappa}{\|\mathcal{B}_1\|_{\mathrm{op}}} \quad \implies \quad \|y^* - y\| > \kappa \frac{\gamma}{\|\mathcal{B}_1\|_{\mathrm{op}}} \quad \text{where } \|\cdot\|_{\mathrm{op}} \text{ is the operator norm}$$

*for some constant $\gamma$, if $\|x^* - x\| < \epsilon$. In other words, one can make $\|x^* - x\|$ arbitrarily small while allowing $\|y^* - y\|$ to remain arbitrarily large, due to the interplay between $(c^*, s^*)$ and $(c, s)$.*

Theorem 3.3 shows that even when the distance between $x^*$ and $x$ is made arbitrarily small, the distance between the corresponding $y^*$ and $y$ can still be significant, as there is no upper bound on this gap. Consequently, the predicted value based on $x^*$ may coincide with $y$, causing a discrepancy with the true $y^*$.

Picking demonstrations from $\mathcal{D}_c$ yields better in-context learning than picking from $\mathcal{D}_x$. The upper bound of the estimation error of the learned parameter is smaller compared to that of input-based selection. Furthermore, the upper bound on the test prediction error with CCL is also smaller. The parameter update in ICL, under a transformer architecture, is approximated by gradient descent on the demonstration set, following the formulation in [36]. Let $w_c^{(M)}$ be the weight updated via $M$ steps of gradient descent using the empirical risk on $\mathcal{D}_c$, and let $w_x^{(M)}$ be the corresponding weight updated from $\mathcal{D}_x$.

**Theorem 3.4** (Performance of the $c$-similarity). *For sufficiently large $N, M$, with probability at least $1 - \delta_{tail}$, the following holds under Assumption C.1–C.4 in Appendix C:*

1. ***Tighter upper bound on estimation error.*** *The estimation errors admit upper bounds $U_{param}^c$ and $U_{param}^x$ such that*

$$\|w_c^{(M)} - w^*\| \le U_{param}^c, \quad \|w_x^{(M)} - w^*\| \le U_{param}^x, \quad \text{and} \quad U_{param}^c < U_{param}^x.$$

   $U_{param}^c = (1/\lambda_{min}(\Gamma_c)) \cdot C_u S_N$ *and* $U_{param}^x = (1/\lambda_{min}(\Gamma_x)) \cdot C_u S_N$. $C_u$ *is a some constant and* $S_N = \sqrt{\log(1/\delta_{tail})/N}$. $\lambda_{min}(A)$ *denotes the minimum eigenvalue of a matrix $A$. $\Gamma_c$ and $\Gamma_x$ are the empirical second moment matrices of $\mathcal{D}_c$ and $\mathcal{D}_x$.*

2. ***Tighter upper bound on test error.*** *For the test query $(x^*, y^*)$ with $y^* = (w^*)^\top c^* + \varepsilon_y^*$, the prediction errors admit upper bounds $U_{test}^c$ and $U_{test}^x$ such that*

$$\left|(w_c^{(M)})^\top x^* - y^*\right| \le U_{test}^c, \quad \left|(w_x^{(M)})^\top x^* - y^*\right| \le U_{test}^x, \quad \text{and} \quad U_{test}^c < U_{test}^x.$$

   $U_{test}^c = \|x^*\| U_{param}^c + |\mathcal{R}|$ *and* $U_{test}^x = \|x^*\| U_{param}^x + |\mathcal{R}|$. $\mathcal{R} = (w^*)^\top x^* - y^*$.

Theorem 3.4 shows that, with high probability, the parameter error $\|w_c^{(M)} - w^*\|$ and the test error $\|(w_c^{(M)})^\top x^* - y^*\|$ admit upper bounds that are tighter than the corresponding bounds obtained from $\mathcal{D}_x$. In essence, when $\mathcal{D}_c$ is used for demonstrations, the underlying design matrix becomes better conditioned with respect to $c$, mitigating the confounding effect of $s$ and leading to tighter error bounds.

## 4 Experiments

We validate the effectiveness and validity of CCL by addressing three main points. First, in Section 4.2, we verify that the latent variables $c$ and $s$ inferred by CCL indeed capture domain-invariant and domain-variant features, respectively, for modeling the causal factors of $x$. In Section 4.3, we examine whether the samples characterized by $c$ exhibit similarity to the test samples or convey the same underlying intent. In Section 4.4, we evaluate how the demonstration sets constructed using CCL enhance in-context learning performance under OOD scenarios. In Section 4.5, we qualitatively analyze how the latent features $c$ and $s$ capture distinct features. Lastly, in Section 4.6, we investigate the capability of CCL on new or more intricate reasoning tasks and perform a sensitivity analysis.

### 4.1 Experimental setup

We adopt a query-dependent demonstration strategy that dynamically selects the suitable examples for each test input. After embedding a test query, we compute its cosine distances to all candidates in the in-distribution training pool. In the K-nearest-neighbor (KNN) variant, the $K$ closest instances, where $K$ equals the predefined shot size ($\Omega$), are selected directly. We also investigate a K-means-based selection method that is governed by two hyperparameters, $R$ and $P$. A proportion $R$ of the shot budget is allocated to the most similar instances, obtained exactly as in the KNN procedure. The remaining budget $K = \Omega - R$ is filled by clustering: among the next $P$ (with $P \in \{50, 100, 300\}$) most similar candidates, we run K-means clustering and, from each cluster, select the sample whose embedding is closest to the centroid. This combined strategy yields prompts that simultaneously maintain high relevance to the query while covering a broader range of semantic regions.

### 4.2 Synthetic data

| Method | ID Task Comparison | | | Env. Comparison | | | OOD Task Comparison | | |
|---|---|---|---|---|---|---|---|---|---|
| | Acc. | NDCG | F1 | Acc | NDCG | F1 | Acc | NDCG | F1 |
| x | 57.7 | 71.5 | 58.6 | 85.7 | 91.3 | 86.0 | 45.0 | 57.9 | 45.4 |
| CVAE ($z$) | 33.3 | 60.2 | 32.5 | 33.1 | 43.2 | 32.2 | 32.4 | 53.6 | 33.9 |
| Oracle ($c$) | **100.0** | **100.0** | **100.0** | 33.5 | 48.7 | 33.7 | **100.0** | **100.0** | **100.0** |
| CCL ($c$) | **100.0** | **100.0** | **100.0** | 40.8 | 55.0 | 39.9 | **100.0** | **100.0** | **100.0** |
| Oracle ($s$) | 33.3 | 48.9 | 33.9 | **100.0** | **100.0** | **100.0** | 32.7 | 51.3 | 33.0 |
| CCL ($s$) | 36.2 | 51.4 | 36.2 | **100.0** | **100.0** | **100.0** | 33.2 | 48.3 | 31.9 |

Table 1: Retrieval experiments on synthetic data show that CCL consistently outperforms alternatives on both in-distribution and out-of-distribution task queries, confirming that $c$ captures the underlying causal structure of the tasks. Conversely, when retrieval is conditioned on environment labels, $s$-based retrieval excels, highlighting their sensitivity to domain-specific factors. CCL's learned representation, CCL ($c$), tracks the ground-truth causal feature particularly closely.

We construct synthetic data with three tasks and five environments. Following Figure 2b, we first define the root nodes: the task variable $t$ and the $s$ variable. We enforce independence among task embeddings $t$ by randomly initializing them with orthogonality constraints, applying the same approach to $s$. Then, we generate the $c$ embedding using a three-layer fully connected neural network that takes $t$ as input and add random noise to its output. Other variables follow a similar process. We train the neural networks, viewed as non-linear data-generating functions, using contrastive learning to ensure that $c$ is similar within the same task and $e$ is similar within the same $s$, while enforcing dissimilarity across different tasks or environments.

To better reflect realistic scenarios, we consider similar tasks or environments. Specifically, for the root nodes $t$ and $s$, we set the cosine similarity between any two $t$ or $s$ embeddings to a value between 0 and 1 (in our experiment, we use 0.7). During contrastive training of the generating functions,

we adjust the loss weights to reduce the penalty for similar tasks or environments, ensuring their embeddings are not pushed too far apart.

Table 1 presents the proportion of retrieved samples whose task or environment (Env.) matches that of the target input, across different embedding types, under both in-distribution (ID) and out-of-distribution (OOD) settings. Additional experimental results and discussions on the synthetic experiments are provided in Appendix D.

## 4.3 MGSM

| Metric | $x$ embedding | $c$ embedding |
|---|---|---|
| Total Accuracy | 81.03 | **85.84** |
| ID Accuracy | 97.05 | **99.74** |
| OOD Accuracy | 53.00 | **61.52** |
| Total NDCG | 86.00 | **88.73** |
| ID NDCG | 99.12 | **99.89** |
| OOD NDCG | 63.03 | **69.21** |

(a) Comparison of retrieval accuracy and NDCG for $x$ and $c$ embeddings on MGSM in the 5-shot setting.

| Method | Total | ID | OOD |
|---|---|---|---|
| ZS | 87.71 | 89.43 | 84.70 |
| ICL (Fix.) | 91.20 | 91.26 | 91.10 |
| ICL (KNN) | 94.07 | 95.83 | 91.00 |
| CCL | **94.55** | **96.11** | **91.80** |

(b) Comparison of performance. ZS denotes the zero-shot baseline, ICL (Fix.) uses a fixed demonstration set. ICL (KNN) and CCL utilize KNN retrieval

Table 2: (a) compares five-shot MGSM retrieval performance between embeddings derived from the original inputs $x$ and from the causal features, $c$. (b) reports overall, in-distribution (ID), and out-of-distribution (OOD) accuracies for four prompting regimes—zero-shot (ZS), fixed demonstrations, KNN-based retrieval, and CCL.

As another dataset to evaluate the performance of our methodology, we employ the MGSM (Multilingual Grade School Math) dataset [37]. The MGSM dataset is a human-annotated translation of 250 problems from the GSM8K dataset [38] into ten different languages.

Utilizing the MGSM dataset, our goal is to evaluate the precision with which CCL deduces latent variables $c$, that represent the fundamental context of problems. For this purpose, we evaluate the retrieval performance by examining how correctly the model retrieves the same problem given a specific question.

First, we extract embeddings for each question using OpenAI's text-embedding-3-small model. Based on these embeddings, we split the data into an ID and an OOD dataset. We use Swahili, Thai, Telugu, and Bengali for the OOD dataset, while the remaining languages are designated as ID. We provide a detailed explanation of the classification criteria in Appendix D.

In this experiment, we define the problem category as the task $t$. The categories include six classes, such as "Arithmetic Operations" and "Geometry and Measurements". These categories are generated by labeling each question using OpenAI's o1, followed by human verification. During the labeling process, only English questions are labeled, and the same labels are directly applied to corresponding questions in other languages.

Table 2a presents the retrieval performance of the $x$ embeddings and the $c$ embeddings. We evaluate how accurately each method retrieves the same problem in a different language. The results demonstrate a significant improvement in accuracy and NDCG for both ID and OOD when using our approach instead of $x$ embeddings.

Next, we perform ICL based on the retrieval results. In the MGSM dataset, we evaluate performance by measuring the model's prediction accuracy. Similarly to the retrieval process, we use a 5-shot setting to assess performance and compare zero-shot (ZS), ICL (Fixed sample, KNN) and CCL. Unlike ICL (KNN) and CCL, which can retrieve samples from different languages, ICL (Fix.) uses predefined samples specific to each language. We use GPT-4o-mini for in-context learning. We refer to Appendix D for a detailed explanation of the MGSM experiment.

Table 2b illustrates the experimental results. The results demonstrate that CCL-based retrieval for in-context samples achieves higher accuracy in both ID and OOD settings than other approaches. This aligns with the strong retrieval performance of $c$ embedding indicated in Table 2a, demonstrating that selecting in-context samples based on the latent causal feature $c$ is crucial for problem solving and improves in-context learning accuracy.

## 4.4 Generalization across tasks and domains

| Language model | Retrieval method | QNLI | PIQA | WSC273 | YELP | Avg. |
|---|---|---|---|---|---|---|
| Llama-3.2-3B-IT | ZS | 43.36 | **71.33** | 55.31 | *88.98* | 64.75 |
| | LLM-R | 29.93 | 69.91 | 61.17 | 79.48 | 60.12 |
| | ICL (K-means) | 68.13 | 69.04 | 49.82 | 75.81 | *65.70* |
| | CCL | **75.18** | *70.46* | **61.91** | **95.44** | **75.74** |
| Phi-4-mini-IT | ZS | **86.34** | **76.01** | 64.10 | 95.76 | 80.55 |
| | LLM-R | *85.21* | 74.10 | 65.93 | **96.37** | 80.40 |
| | ICL (K-means) | 83.18 | 74.81 | *71.06* | 96.25 | *81.33* |
| | CCL | 82.26 | *75.73* | **71.43** | *96.33* | **81.44** |
| GPT-4o | ZS | **91.30** | *94.07* | 90.84 | 97.47 | 93.42 |
| | LLM-R | 90.32 | **94.23** | *92.67* | *98.27* | 93.87 |
| | ICL (K-means) | 88.28 | 93.04 | 87.55 | 98.17 | 91.76 |
| | CCL | *90.77* | 93.15 | **93.77** | **98.36** | **94.01** |

Table 3: Out-of-distribution accuracy on QNLI, PIQA, WSC273, and Yelp for three language models—Llama-3.2-3B-IT, Phi-4-mini-IT, and GPT-4o—under four prompting regimes: zero-shot (ZS), the learned-retriever baseline (LLM-R), and two K-means-based retrieval approaches, vanilla ICL and CCL. Bold numbers denote the highest score in each column, and italics denote the second highest. CCL attains the best average accuracy for every model, with particularly pronounced improvements for the smaller Llama-3.2-3B-IT.

We evaluate whether examples selected by CCL improve performance on OOD NLP tasks. Adopting the experimental protocol of LLM-R [39], we compare against their retrieval method but instead assess the generated outputs rather than relying on token probabilities. Our approach retrieves examples with similar $c$ embeddings via KNN, clusters them using K-means, and selects the cluster centers as final candidates. As shown in Table 3, CCL consistently yields strong performance across diverse OOD tasks. We follow the same 8-shot setting used in LLM-R to ensure a fair comparison.

### 4.4.1 Sensitivity to the embedding models

| Language model | Embedding model | QNLI | PIQA | WSC273 | YELP | Avg. |
|---|---|---|---|---|---|---|
| Phi-4-mini-IT | text-embedding-3-small | **82.26** | **75.73** | *71.43* | **96.33** | *81.44* |
| | multilingual-e5-large-instruct | **82.26** | *75.25* | **73.99** | *95.72* | **81.81** |

Table 4: CCL accuracy on four out-of-distribution benchmarks when the same language model (Phi-4-mini-IT) is paired with two embedding models (OpenAI's text-embedding-3-small and the multilingual-e5-large-instruct). Scores are given for each task and averaged; bold indicates the highest score per column, and italics the second-highest. The multilingual-e5 encoder attains the top overall score, yet the gap is small, indicating that CCL remains robust to the choice of embedding model.

To evaluate CCL's sensitivity to the encoder, we reran the entire pipeline across the NLP benchmarks using multilingual-e5-large-instruct [40], an open-source embedding model that ranks among the top performers on the MTEB text-embedding leaderboard [41]. Table 4 experimentally demonstrates that CCL maintains comparable performance despite changes in the embedding model, highlighting its robustness in inferring causal features.

## 4.5 Qualitative analysis

We provide a qualitative analysis of the learned latent features to better understand how $c$ and $s$ are interpreted in practice. To visualize the semantics encoded in these variables, we decode sentence embeddings while zeroing out one latent dimension. Specifically, we first infer $c$ and $s$ from an input embedding $x$. We then set $s = 0$ to generate $x'_{s=0}$, which highlights the domain-invariant features represented by $c$. Similarly, we set $c = 0$ to generate $x'_{c=0}$, which reveals the domain-variant information captured by $s$. Table 5 lists the top-5 nearest words to each decoded embedding.

| $x$ | $x'_{s=0}$ | $x'_{c=0}$ |
|---|---|---|
| horribleappetizers | unappetizing | review |
| pancakes | flavorless | reviewers |
| potatos | horribleappetizers | critiques |
| hadhorrible | inedible | soggy |
| bad | trashed | reviews |

(a) Original negative sentence is "*the red velvet pancakes were horrible and brown, and potatos were over cooked and bland.. would not recommend*"

| $x$ | $x'_{s=0}$ | $x'_{c=0}$ |
|---|---|---|
| dvd | unusable | reverb |
| eject | expired | throw |
| disks | cancelled | film |
| unusable | crappy | review |
| purchased | trashed | trip |

(b) Original negative sentence is "*Worked for about 4 months. DVD player will not eject or accept disks. Do not buy.*"

Table 5: Top-5 nearest words on Yelp and Amazon. The sentence embedding $x$ captures both semantic and contextual tokens. In contrast, $x'_{s=0}$ clusters strongly around negative sentiment expressions, while $x'_{c=0}$ clusters tokens associated with contextual metadata.

## 4.6 Generalization and sensitivity analysis

### 4.6.1 Advanced tasks

Table 6 presents the generalization capability of CCL across advanced tasks. The unseen generation task involves sentiment reversal paraphrasing: the model rewrites a negative sentence to express the opposite sentiment, and we automatically assess its sentiment using GPT-4o-mini. Although CCL trains only on classification tasks, it generalizes well to this unseen generation setting.

|  | Unseen & generation | | Reasoning | Multi-hop QA |
|---|---|---|---|---|
|  | Yelp | Amazon | MMLU | HotpotQA |
| ZS | 86.26 | 86.73 | 60.48 | 82.43 |
| ICL | 87.68 | 85.80 | 61.37 | 84.14 |
| CCL | **90.05** | **87.70** | **61.52** | **84.43** |

Table 6: Performance comparison of ZS, ICL, and CCL across tasks using Phi-4-mini-IT.

For MMLU [42], we retrieve five examples for each query without distinguishing among the 57 domains. For HotpotQA [43], we provide each query with its corresponding document and retrieve examples to form document-example pairs. This experiment provides evidence that CCL may help with hierarchical and composite language-understanding problems.

### 4.6.2 Sensitivity analysis

|  | QNLI | PIQA | WSC273 | YELP |
|---|---|---|---|---|
| ZS | 43.4 (± 0.00) | 71.3 (± 0.00) | 55.3 (± 0.00) | 89.0 (± 0.00) |
| LLM_R | 29.9 (± 0.00) | 69.9 (± 0.00) | 61.2 (± 0.00) | 79.5 (± 0.00) |
| ICL | 68.1 (± 0.00) | 69.0 (± 0.00) | 49.8 (± 0.00) | 75.8 (± 0.00) |
| CCL | **75.2 (± 0.45)** | **72.4 (± 1.12)** | **58.98 (± 2.78)** | **95.10 (± 0.25)** |

(a) Mean accuracy and std over 5 random seeds.

| dim($c$) | QNLI | PIQA | WSC273 | YELP | Avg. |
|---|---|---|---|---|---|
| 128 | 69.5 | **72.6** | 56.1 | 93.5 | 72.9 |
| 256 | **75.3** | 71.6 | 60.4 | 94.9 | 75.6 |
| 1024 (ours) | 75.2 | 70.5 | **61.9** | **95.4** | **75.7** |

(b) Accuracy variation w.r.t. dim($c$).

Table 7: Performance of CCL under different training conditions using Llama-3.2-3B-IT. (a) OOD benchmark accuracy across five random seeds, showing stable results despite stochastic variation in VAE training. (b) Performance changes with respect to latent dimensions, indicating that smaller dimensions do not significantly degrade accuracy.

Table 7a shows the OOD benchmark results under different random seeds used for training the VAE within CCL. Since response generation is deterministic (non-sampling), other baselines exhibit zero variance. Table 7b reports the effect of varying the latent dimensions of $c$ and $s$ during VAE training. The results suggest that model performance remains stable even with smaller latent dimensions.

## 5 Conclusion and discussion

We propose CCL, the first framework to integrate causal representation learning into ICL, addressing a key limitation of conventional ICL in OOD settings. By selecting demonstrations based on causal representation rather than surface-level similarity, CCL improves robustness, and parameter estimation, with theoretical guarantees.

**Limitation and Impact statement.** Since CCL employs a VAE-based latent embedding, the inherent structural limitations of VAE may hinder its ability to fully capture the rich and nuanced representations of natural language. We leave the deeper integration of embedding-based retrieval with causal inference as future work.

## Acknowledgments

This work was supported by the National Research Foundation of Korea(NRF) grant funded by the Korea government(MSIT)(RS-2024-00457216).

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
