# OpenReview forum: "CCL: Causal-aware In-context Learning for Out-of-Distribution Generalization"
_NeurIPS.cc/2025/Conference — NeurIPS 2025 poster_

### Official Review · Reviewer_1SuJ · 2025-06-23

**Clarity:** 3
**Significance:** 3
**Originality:** 3
**Rating:** 4
**Confidence:** 3

**Summary:**

This paper proposes Causal-aware In-Context Learning (CCL) to address the performance limitations of traditional in-context learning (ICL) under out-of-distribution (OOD) settings. The authors argue that conventional ICL methods, which rely on superficial similarity in input xxx, often fail when demonstration samples come from different semantic domains despite having structurally similar forms. To remedy this, the paper leverages causal representation learning, using a VAE-based framework to extract task-relevant latent variables c that remain invariant across environments, while separating out spurious factors s. CCL selects demonstrations based on similarity in ccc, rather than x, leading to improved OOD generalization. Both theoretical analysis and empirical results confirm that demonstrations can diverge from the surface form of x as long as the semantic (causal) alignment is preserved.

**Questions:**

1.	Can provided an example of x, c and s? In particular, is the modeling of s strictly necessary for the success of CCL, or could a simplified formulation using only c suffice for identifying useful demonstrations?

2.	How you select x for each test query for generating c and s? Is there a separate VAE trained per task/domain? How is the demonstration pool determined for a new query during test time—does it rely on fixed candidate sets from the source distribution?

3.	In Table 4, CCL underperforms on some tasks. Have the authors conducted any error analysis to investigate these cases? Is there evidence that these failures stem from incorrect estimation of the causal variables c?

4.	More broadly, could the authors discuss the identifiability of the causal representations under their VAE framework and whether the observed performance drop is due to limitations in causal inference?

**Ethical Concerns:**

["NO or VERY MINOR ethics concerns only"]

**Final Justification:**

The clarification resolved my questions. I keep my positive review for this paper.

**Limitations:**

yes

**Quality:**

3

**Strengths And Weaknesses:**

Strength

•1.  Novelty & Significance: The paper introduces a principled and theoretically grounded method for demonstration selection in ICL by emphasizing causal similarity. This represents a meaningful step forward for robust ICL under distribution shift.

•2.  Theoretical Depth: The method is supported by a rigorous theoretical framework. The authors clearly define their assumptions and provide formal guarantees demonstrating the benefits of selecting demonstrations based on causal representations. This theoretical grounding makes the proposed approach compelling and credible.•

3.  Comprehensive Experiments: Experiments on both synthetic and real-world datasets demonstrate the effectiveness of the method. The design clearly isolates and evaluates the role of causal representations, validating the method's robustness in OOD scenarios.

Weakness:
See Questions

---

> ### Author Rebuttal · Authors · 2025-07-31
>
> Thank you very much for your detailed and insightful review. Your feedback has been instrumental in helping us refine and improve the clarity and robustness of our work. We have carefully considered each of your comments. We hope that our responses below effectively clarify the concerns, and we remain open to providing further clarification where needed.
>
> > Can provided an example of x, c and s?
>
> Thank you for this important question. We illustrate how these latent variables behave in practice using a qualitative decoding analysis from our trained VAE under the CCL framework.
>
> To assess the interpretability of the latent variables, we first infer $c$ and $s$ from a given input sentence embedding $x$, and then decode variants of the input by zeroing out either $c$ or $s$
> - $x_{s=0}'$ retains only the information encoded in $c$ (task-relevant).
> - $x_{c=0}'$ retains only the information encoded in $s$ (domain-specific).
>
> Because $x_{s=0}'$ and $x_{c=0}'$ lie in the same embedding space as $x$, we retrieved nearest neighbors to reveal the types of semantic content each latent variable preserves.
>
> Qualitative Example 1 (Yelp dataset):
> Original negative sentence: “the red velvet pancakes were horrible and brown, and potatos were over cooked and bland.. would not recommend”
>
> Table 1. Top 5 nearest words
>
> | $x$                | $x_{s=0}'$         | $x_{c=0}'$ |
> | --------           | --------           | --------   |
> | horribleappetizers | unappetizing       | review     |
> | pancakes           | flavorless         | reviewers  |
> | potatos            | horribleappetizers | critiques  |
> | hadhorrible        | inedible           | soggy      |
> | bad                | trashed            | reviews    |
>
> In Table 1,
> - $x$ retrieves both semantic and domain-specific tokens (e.g., “pancakes”, “potatos”).
> - $x_{s=0}'$ clusters around sentiment-related tokens, e.g., “unappetizing”, “inedible”.
> - $x_{c=0}'$ surfaces meta-review terms such as “review”, “critique”, which likely reflect domain-specific features.
>
> Qualitative Example 2 (Amazon dataset):
> Original negative sentence: “Worked for about 4 months. DVD player will not eject or accept disks. Do not buy.”
>
> Table 2. Top 5 nearest words
> | $x$       | $x_{s=0}'$  | $x_{c=0}'$ |
> | -  | -   | - |
> | dvd       | unusable    | reverb   |
> | eject     | expired     | throw    |
> | disks     | cancelled   | film     |
> | unusable  | crappy      | review   |
> | purchased | trashed     | trip     |
>
> Similarly, in Table 2,
> - $x_{s=0}’$ captures failure-related sentiment (e.g., “unusable”, “trashed”).
> - $x_{c=0}’$ retrieves words like “film” and “review”, showing latent connection to domain/style.
>
> These examples highlight that the CCL is able to effectively separate task-relevant and task-irrelevant factors across domains.
>
> > In particular, is the modeling of s strictly necessary for the success of CCL, or could a simplified formulation using only c suffice for identifying useful demonstrations?
>
> - The reason we explicitly model two latent variables, $c$ and $s$, is to isolate domain-invariant features ($c$) for the purpose of robust retrieval particularly in OOD settings.
> - If we were to use a single latent variable ($z$) without decomposing it into $c$ and $s$, then $z$ would have to represent both the invariant and variant factors embedded in $x$. This conflation leads to two key problems:
>   - (1) Entangled representation: without disentanglement, the latent variable may capture spurious correlations tied to specific domains or datasets. Consequently, retrieval based on $z$ may select examples that are similar on superficial terms (e.g., domain, writing style), rather than the underlying invariant feature.
>   - (2) Inconsistent behavior across domains: Since s is highly sensitive to environmental factors (e.g., dataset source), a $z$ that mixes $c$ and $s$ will behave inconsistently under distribution shifts, undermining the core motivation of CCL — OOD generalization.
> - This point is reinforced both theoretically (Section 3.4) and empirically. Theorems 3.3 and 3.4 show that selecting demonstrations based on $x$ (or entangled $z$) can lead to large prediction errors due to confounding from $s$, whereas focusing on $c$ leads to improved convergence and lower test error.
> - In synthetic experiments (Table 1 in the paper), retrieval based on $c$ clearly outperforms that based on $x$ or $s$ when the goal is to retrieve task-relevant demonstrations, especially in OOD scenarios. This confirms that disentangling is not only theoretically desirable but necessary in practice to ensure that $c$ learns the invariant features.
>
> > How you select x for each test query for generating c and s? Is there a separate VAE trained per task/domain?
>
> We appreciate the opportunity to clarify this point. The VAE used in CCL is trained on a collection of benchmark datasets spanning a wide range of tasks and domains. In other words, a single shared VAE is trained using data from diverse tasks and domains, enabling it to generalize and infer latent representations ($c$, $s$) across heterogeneous inputs.
>
> At test time, for each test query $x_t$, we compute its embedding and pass it through the pretrained VAE to infer its corresponding latent causal variable $c_t$ and spurious variable $s_t$. No task- or domain-specific retraining of the VAE is required.
>
> > How is the demonstration pool determined for a new query during test time—does it rely on fixed candidate sets from the source distribution?
>
> CCL is specifically designed to be robust in out-of-distribution (OOD) settings. During inference, we assume access to a fixed and task-agnostic demonstration pool, which does not include examples from the same distribution as the test query.
>
> Given a new test query $x_t$, we first extract its sentence embedding and use the pretrained VAE to infer $c_t$. Then, using similarity in the $c$-space, we retrieve the nearest examples from the fixed pool to construct a demonstration prompt. This retrieval process is entirely independent of task-specific labels or distributions.
>
> > In Table 4, CCL underperforms on some tasks. Have the authors conducted any error analysis to investigate these cases?
>
> Thank you for raising this important point. We conducted an error analysis to explore the correlation between model performance and the quality of $c$ inferred by the VAE.
>
> Specifically, we measured the average reconstruction loss of both input $x$ and output $y$ on in-distribution (ID) and out-of-distribution (OOD) datasets across four task types:
> - Coreference resolution: ID = WSC, OOD = WSC273
> - Natural language inference: ID = RTE, OOD = QNLI
> - Sentiment analysis: ID = Sentiment140, OOD = Yelp
> - Commonsense reasoning: ID = COPA, OOD = PIQA
>
> For each dataset, we calculated the reconstruction loss on the test set using the pretrained VAE.
>
> Interestingly, we found that the commonsense reasoning tasks (COPA/PIQA) exhibited the highest reconstruction loss, particularly on the output $y$. This suggests that the VAE had greater difficulty modeling causal representations in this domain, potentially due to the more abstract or diverse semantics involved.
>
> These findings imply that weaker performance on certain tasks (e.g., PIQA) may indeed stem from less accurate inference of $c$, highlighting a valuable direction for improving the robustness of causal variable estimation.
>
> Importantly, we appreciate the reviewer’s insight, which has led us to a new line of investigation. In our current setup, we balanced the training dataset sizes across domains. However, we now realize that equal data volume may not translate to equal modeling difficulty across tasks.
>
> > More broadly, could the authors discuss the identifiability of the causal representations under their VAE framework and whether the observed performance drop is due to limitations in causal inference?
>
> Thank you for raising this important question. We would like to address your concern from both a conceptual and empirical perspective.
>
> - Our approach is inspired by works such as CEVAE [1], where a VAE is used to estimate the effect of unobserved confounders on treatment and outcome variables. Importantly, in such settings, the goal is not to explicitly recover the exact identity of the latent confounder, but rather to learn a latent variable that captures its functional influence on treatment and outcome.
> - Similarly, our framework does not aim to provide full identifiability of the latent causal variable $c$, but instead learns a functional approximation that is sufficient for improved generalization under distribution shift.
> - In line with this, our VAE-based model learns two latent variables — $c$ (domain-invariant, task-relevant) and $s$ (domain-variant) — whose separation is functionally meaningful. This is supported empirically:
>   - In synthetic experiments (Table 1 in the manuscript), we show that $c$ and $s$ align with ground-truth task and environment variables.
>   - In real data (Tables 1 and 2), decoding from $c$ and $s$ yields clearly different semantics, showing the model effectively separates domain-invariant and domain-variant information.
> - However, as mentioned earlier, we believe that the performance drop in certain tasks is less about the disentanglement between $c$ and $s$, and more about the expressiveness or optimization of $c$ itself in predicting $y$.
> - A promising direction is to enhance the VAE’s representation capacity. The current VAE in CCL has only 6.3M parameters (~0.2% the size of LLaMA 3.2 3B). A larger or more expressive VAE could lead to better modeling of $c$.
> - Your review motivated us to explore the correlation between the reconstruction loss of $y$ (from $c$) and NLP task performance. This suggests a new direction for improving the robustness of CCL by better optimizing the latent causal representation $c$. We will include this analysis and discussion in the revision.
>
> [1] Louizos et al., Causal Effect Inference with Deep Latent-Variable Models, NeurIPS 2017

---

> > ### Comment · Reviewer_1SuJ · 2025-08-04
> >
> > Thank you for the clarification and they resolve my questions. I keep my positive review for this paper.

---

> > > ### Author Response · Authors · 2025-08-08
> > >
> > > Thank you for your kind response and for maintaining your positive review of our paper.

---

### Official Review · Reviewer_2GL1 · 2025-06-28

**Clarity:** 2
**Significance:** 2
**Originality:** 2
**Rating:** 3
**Confidence:** 4

**Summary:**

This paper introduces a new method called Causal-aware In-context Learning (CCL) to improve the performance of large language models (LLMs) on out-of-distribution (OOD) datasets, where traditional in-context learning (ICL) methods fall short. The proposed CCL method is inspired by causal representation learning and aims to select better demonstration sets for LLMs by focusing on the underlying causal relationships in the data rather than superficial features. To achieve this, CCL uses a novel VAE-based causal representation learning technique to identify and capture these causal features. The paper provides both theoretical and empirical evidence that CCL improves the OOD generalization performance of LLMs by selecting demonstration sets that are causally related to the target query. By doing so, CCL helps LLMs to generalize better even in unseen environments.

**Questions:**

N/A

**Ethical Concerns:**

["NO or VERY MINOR ethics concerns only"]

**Final Justification:**

I appreciate the analyses, but my concern remains: a single‐layer VAE—even with dual reconstruction—still oversimplifies language’s deeply entangled, context-dependent causal factors. Without broader benchmarks or formal guarantees, it’s unclear whether CCL truly learns causal structure rather than surface correlations.

My score keeps unchanged.

**Limitations:**

Yes

**Quality:**

2

**Strengths And Weaknesses:**

Strengths:
1. The methodology is designed to prevent the LLM from relying on misleading "shortcuts" or spurious correlations in the data. By forcing the model to consider the causal structure, CCL encourages a deeper understanding of the task, which in turn leads to better performance.

Weakness:
1. The process of learning causal representations with a VAE and then using this model to select demonstrations for every new query could be computationally expensive. The paper may not adequately address the practical scalability of this approach to very large models or datasets, which could be a significant barrier to real-world application.
2. The method assumes that causal and non-causal features can be cleanly separated into distinct latent variables. In practice, especially in complex domains like natural language, these factors might be heavily entangled, making a clean separation difficult and potentially limiting the effectiveness of the approach.
3. The theoretical model might oversimplify the nature of causality in language. It assumes a relatively straightforward mapping from latent variables to observed data, which may not capture the intricate, context-dependent, and multi-layered causal relationships inherent in language and reasoning.

---

> ### Author Rebuttal · Authors · 2025-07-31
>
> We sincerely thank you for the constructive and forward-looking feedback, which has been very helpful in guiding improvements to our work. I hope the responses below sufficiently address your concerns and questions. If anything remains unclear or requires further clarification, we would be more than happy to provide additional explanation.
>
> > The process of learning causal representations with a VAE and then using this model to select demonstrations for every new query could be computationally expensive. The paper may not adequately address the practical scalability of this approach to very large models or datasets, which could be a significant barrier to real-world application.
>
> - We appreciate this insightful comment. Many retrieval-based ICL methods — such as LLM-R [1], TTF [2] — train a retriever to find optimal demonstrations for a given input. These approaches typically rely on open-source language models as retrievers, requiring access to the full or partial model weights during training and inference.
>
> - In contrast, CCL only requires a relatively lightweight VAE for training, and its latent variables ($c$, $s$) are significantly more compact than raw text embeddings. The parameter size of our VAE is 6.3M, which is approximately 0.2% of the smallest language model we evaluated (LLaMA 3.2 3B Instruct).
>
> - To quantify the computational overhead, we compared CCL with embedding-based ICL without any retriever training. The average time to extract embeddings using OpenAI’s text-embedding-3-small API is about 0.416s per sample, while the multilingual-e5-large-instruct open-source model takes about 0.032s.
>
> - For CCL:
>   - VAE training takes approximately 20.4 minutes across 5 random seeds.
>   - A single training iteration for one sample takes 0.06s.
>   - Inference time for generating $c$ and $s$ embeddings is 0.003s per sample.
>   - Construction of an 8-shot prompt via k-means clustering takes about 0.2s, same as ICL.
>   - Inference using LLaMA 3.2B IT on a single A6000 GPU shows average response times of:
>      - CCL: 0.10s
>      - ICL: 0.16s
>
> - Thus, the total time per sample is 0.78s for CCL vs. 0.63s for ICL — a marginal increase. In our view, when the VAE is pre-trained and $c$ embeddings are pre-computed for the example pool, the inference-time overhead could be sufficiently small to make CCL practically usable in real-world scenarios.
>
> > The method assumes that causal and non-causal features can be cleanly separated into distinct latent variables. In practice, especially in complex domains like natural language, these factors might be heavily entangled, making a clean separation difficult and potentially limiting the effectiveness of the approach.
>
> - We agree that in natural language, the disentanglement of causal and non-causal factors can be inherently challenging due to semantic and contextual entanglement. To explore this concern, we performed a qualitative analysis of the learned latent features to better understand how $c$ and $s$ are interpreted in practice.
>
> - As mentioned in the paper, our objective is to decompose each sentence into components relevant to task objectives (domain-invariant) and those that are not (domain-variant). For instance, in sentiment analysis:
>   - Domain-invariant features capture polarity (positive/negative sentiment).
>   - Domain-variant features depend on dataset characteristics — e.g., service or food for Yelp, product quality or usability for Amazon.
>
> - To visualize the meaning encoded in $c$ and $s$, we decoded sentence embeddings with one latent dimension zeroed out. Specifically:
>   1. Infer $c$ and $s$ from input embedding $x$.
>   2. Set $s=0$ to generate $x_{s=0}'$ (highlighting $c$).
>   3. Set $c=0$ to generate $x_{c=0}'$ (highlighting $s$).
>   4. Find nearest words to each decoded embedding to interpret feature semantics.
>
> Qualitative example (Yelp): “the red velvet pancakes were horrible and brown, and potatos were over cooked and bland.. would not recommend”
>
> Table 1. Top 5 nearest words
>
> | $x$                | $x_{s=0}'$         | $x_{c=0}'$ |
> | --------           | --------           | --------   |
> | horribleappetizers | unappetizing       | review     |
> | pancakes           | flavorless         | reviewers  |
> | potatos            | horribleappetizers | critiques  |
> | hadhorrible        | inedible           | soggy      |
> | bad                | trashed            | reviews    |
> | | | |
>
> - The original embedding ($x$) captures both semantic and contextual tokens (e.g., “pancakes”, “potatos”). In contrast, $x_{s=0}'$ clusters strongly around negative sentiment expressions, while $x_{c=0}'$ clusters around meta-content like “review”, which aligns with dataset-specific artifacts.
>
> Qualitative Example 2 (Amazon dataset):
> Original negative sentence: “Worked for about 4 months. DVD player will not eject or accept disks. Do not buy.”
>
> Table 2. Top 5 nearest words
> | $x$       | $x_{s=0}'$  | $x_{c=0}'$ |
> | --------  | --------    | -------- |
> | dvd       | unusable    | reverb   |
> | eject     | expired     | throw    |
> | disks     | cancelled   | film     |
> | unusable  | crappy      | review   |
> | purchased | trashed     | trip     |
> | |  |  |
>
> - Similarly, in Table 2, $x_{s=0}'$ captures negative sentiment towards product usability, while $x_{c=0}'$ retrieves terms more loosely associated with contextual metadata.
>
> - These observations suggest that CCL learns interpretable and partially disentangled representations. While a perfect separation may not always be attainable in natural language, the model demonstrates meaningful structuring of causal factors.
>
> > The theoretical model might oversimplify the nature of causality in language. It assumes a relatively straightforward mapping from latent variables to observed data, which may not capture the intricate, context-dependent, and multi-layered causal relationships inherent in language and reasoning.
>
> Thank you for raising this important point. We agree that causal reasoning in natural language is inherently complex and often multi-layered. However, recent advancements in **text embedding models** — many of which are widely used in real-world applications like retrieval-augmented generation (RAG) — suggest that **rich semantic and structural information** can be captured within a **single embedding**. This strengthens the feasibility of learning informative latent variables like $c$ from natural language inputs.
>
> - In our model, the **latent causal variable $c$ is trained not only to help reconstruct the input $x$, but also the output $y$**. This dual reconstruction objective encourages $c$ to **encode information necessary for solving the downstream task**, thereby aligning the learned representation with the task-specific causal features.
>
> - As discussed earlier and demonstrated in Table 1 and Table 2, the VAE is capable of meaningfully disentangling $c$ (domain-invariant) from $s$ (domain-variant). This supports our claim that the simplified causal structure adopted by CCL can still yield interpretable and functionally distinct representations that are aligned with the our goals for ICL in OOD setting.
>
> - Furthermore, we observed a **correlation between the reconstruction quality of the output $y$ and the accuracy on NLP tasks** — suggesting that **improving the expressiveness of $c$ is a potential path to handling tasks that require reasoning about complex causal relationships**.
>
> - Nevertheless, we acknowledge that this **single-layer generative structure may encounter limitations in handling the full complexity of certain reasoning tasks**. In particular, tasks that involve hierarchical or multi-hop reasoning may require richer representations of causality than what our current VAE-based framework offers.
>
> - To address this, we are actively exploring how to **extend CCL beyond the current framework**. Specifically, we are investigating leveraging transformer-based language models, rather than VAEs, to infer latent variables in tasks that require deeper reasoning.
>
> - Transformer architectures are known to implicitly capture hierarchical reasoning structures, and prior work [3] shows that **deeper transformers can handle complex tasks even without explicit chain-of-thought prompting**.
>
> - Moreover, **circuit-based interpretability studies [4] suggest that it is possible to identify which hidden states within transformers contribute most significantly to reasoning processes**. Inspired by these findings, we are exploring approaches to identify and utilize **specific hidden states that encode task-relevant information**, with the goal of **inferring latent causal features such as $c$** more effectively.
>
> This direction represents a potential avenue for enabling CCL to handle more layered causal structures in natural language.
>
> [1] Wang et al., Learning to Retrieve In-Context Examples for Large Language Models, EACL 2024\
> [2] Liu et al., Unraveling the Mechanics of Learning-Based Demonstration Selection for In-Context Learning, ACL 2025\
> [3] Li et al., Chain of Thought Empowers Transformers to Solve Inherently Serial Problems, ICLR 2024\
> [4] Dunefsky et al., Transcoders Find Interpretable LLM Feature Circuits, NeurIPS 2024

---

> ### Comment · Reviewer_2GL1 · 2025-08-05
>
> I appreciate the analyses, but my concern remains: a single‐layer VAE—even with dual reconstruction—still oversimplifies language’s deeply entangled, context-dependent causal factors. Without broader benchmarks or formal guarantees, it’s unclear whether CCL truly learns causal structure rather than surface correlations.

---

> > ### Author Response · Authors · 2025-08-09
> >
> > We appreciate the concern regarding the expressivity of a VAE-based latent space and reconstruction-driven learning when modeling language’s deeply entangled, context-dependent causal factors. **We acknowledge this limitation in the paper and will expand on it.**
> >
> > - Our framework assumes an invariant causal mechanism $p_\theta(y\mid c)$ and the conditional independence $y \perp (x,t,e,s)\mid c$, and our theoretical analysis is developed under a linear-causal approximation.
> >
> > - In the paper, we derive ELBO objective in CCL to reflect the underlying data-generating process and then empirically verify its effectiveness on natural-language tasks. **To examine whether CCL remains beneficial when more complex, multi-step reasoning is required, during the discussion period we additionally evaluated CCL on HotpotQA, a multi-hop QA benchmark that requires integrating evidence across documents via stepwise reasoning.**
> >
> > - The HotpotQA experimental setting is as follows. We first assume that, for each target query, an appropriate document is already provided. This assumption is made because CCL is not a retrieval method like RAG, and we therefore aim to exclude the influence of the retriever.
> >
> > - We further treat each example collected at the $c$-level or $x$-level, together with its corresponding document, as a single shot. The purpose of this setting is to test the expectation that, compared to examples selected solely based on surface-level similarity, providing the model with examples whose questions share similar intent will allow it to learn a more effective information-processing procedure for deriving answers from the given document.
> >
> > Table 3. Accuracy comparison on HotpotQA with Llama-3.2-3B-IT
> > | Method | Acc. |
> > | --- | --- |
> > | ZS | 80.86 |
> > | ICL | 81.00 |
> > | CCL | **82.29** |
> >
> > - **Table 3 shows that CCL yields higher accuracy than zero-shot and ICL on HotpotQA, supporting the effectiveness of CCL for hierarchical or composite language-understanding problems.**
> >
> > - These additional experiments were run during the brief discussion period and are necessarily limited in scope, yet the results are encouraging. In the revised version, we will incorporate today’s clarifications and the HotpotQA findings.
> >
> > Your review not only helped us clarify the direction of this work, but the experiments conducted during the discussion period also suggest clear room to further strengthen our method. We hope this helps address the concerns you raised. Thank you again for your constructive review.

---

### Official Review · Reviewer_Vb3H · 2025-07-02

**Clarity:** 3
**Significance:** 3
**Originality:** 3
**Rating:** 5
**Confidence:** 3

**Summary:**

This paper addresses the brittleness of ICL under distribution shift by learning causal representations of examples. The proposed CCL framework uses a two-branch VAE to disentangle latent causal factors from environment factors, then retrieves demonstrations whose inferred causal factors are closest to those of the test query. Under a linear-causal SCM, the authors prove that selecting by causal similarity yields parameter updates closer to the true causal predictor and strictly lower test error than input-based retrieval.

**Questions:**

1. How does retrieval and ICL accuracy change when $\dim(c)$ is 32 or 128, or when ELBO reconstruction loss varies?
2. Could you provide a ballpark estimate of the computational resources (GPU hours and peak RAM requirements) for VAE training on, say, 100 K examples?
3. Have you applied CCL retrieval to open-ended or chain-of-thought prompts, and what modifications would be needed?

**Ethical Concerns:**

["NO or VERY MINOR ethics concerns only"]

**Final Justification:**

The rebuttal addressed my earlier concerns by adding robustness checks, hyperparameter sensitivity analysis, and preliminary evidence for generalization beyond classification tasks.

### Resolved issues
- Added multi-seed experiments showing stable performance gains across OOD benchmarks, supporting statistical robustness.
- Provided an ablation on latent dimension size, showing that CCL’s performance is not overly sensitive to this parameter.
- Presented an additional experiment on a paraphrasing generation task, demonstrating applicability beyond classification.

### Remaining considerations
- The evaluation still focuses primarily on short-answer tasks; while the generation example is promising, broader testing on complex reasoning or long-form generation would further strengthen generality claims.

Overall, the paper makes a well-supported contribution by integrating causal representation learning into ICL for improved robustness under distribution shift. The rebuttal strengthened both the empirical and theoretical case, and I consider this a strong and relevant contribution to the field.

**Limitations:**

Yes

**Paper Formatting Concerns:**

No concerns

**Quality:**

3

**Strengths And Weaknesses:**

## Strengths
- The paper introduces an approach of integrating causal representation learning into ICL, focusing on robustness to domain shifts and providing a clear motivation for why causal factors should be prioritized over surface-level similarity.
- Under a well-specified linear-causal SCM, the authors offer rigorous theoretical analysis: Theorem 3.3 demonstrates that nearest-neighbor on input can fail arbitrarily badly, while Theorem 3.4 guarantees that causal-based retrieval accelerates convergence to the true predictor and reduces test error.
- Empirically, CCL shows strong performance across diverse settings: it achieves 100% causal match recovery on synthetic OOD splits and improvements on four NLP OOD benchmarks. Moreover, the method is stable to embedding choice, with minimal variation between text-embedding-3-small and multilingual-e5-large.
- The authors provide extensive reproducibility details in the appendix, including full ELBO derivations, proof sketches, synthetic data generation protocols, MGSM split rationale, and exact prompt templates.

---

## Weaknesses
- Reported gains lack any statistical significance testing or confidence intervals, raising concerns about whether smaller improvements are robust or within experimental noise.
- There is no ablation study on the latent representation, such as varying $\dim(c)$ or analyzing the impact of reconstruction quality on retrieval and ICL performance, leaving hyperparameter sensitivity uncharacterized.
- All experiments focus on classification or short-answer tasks, so it remains unknown how CCL would extend to free-form generation or chain-of-thought prompting scenarios.

---

> ### Author Rebuttal · Authors · 2025-07-31
>
> We truly appreciate your suggestions, which have provided valuable directions for strengthening the validity and credibility of our method. We have carefully addressed your comments in the revised version by incorporating the following additional analyses and experiments.
>
> > Reported gains lack any statistical significance testing or confidence intervals, raising concerns about whether smaller improvements are robust or within experimental noise.
>
> - We completely agree that verifying the robustness and reproducibility of our method through repeated trials is critical. In response to your comment, we conducted additional experiments using five different random seeds (\{40, 41, 42, 43, 44\}) to assess the stability of our VAE-based CCL model. The results are presented in Table 1, which reports the mean and standard deviation of CCL’s performance across OOD benchmarks.
> - Note that the LLM was set to perform classification deterministically using greedy decoding to eliminate randomness from the output side. Therefore, other baseline results are not affected by seed variation.
>
> Table 1. Accuracy of OOD benchmarks for Llama 3.2 3B IT
> |        | QNLI           | PIQA            | WSC273         | YELP            |
> | ----   | ---            | ----            | ------         | ----            |
> |  ZS    | 43.4 ($\pm$ 0.00) | 71.3  ($\pm$ 0.00) | 55.3 ($\pm$ 0.00) | 89.0  ($\pm$ 0.00) |
> |  LLM_R | 29.9 ($\pm$ 0.00) | 69.9  ($\pm$ 0.00) | 61.2 ($\pm$ 0.00) | 79.5  ($\pm$ 0.00) |
> |  ICL   | 68.1 ($\pm$ 0.00) | 69.0  ($\pm$ 0.00) | 49.8 ($\pm$ 0.00) | 75.8  ($\pm$ 0.00) |
> | CCL    | **75.2** ($\pm$ 0.45) | **72.4**  ($\pm$ 1.12) |**58.98** ($\pm$ 2.78) | **95.10** ($\pm$ 0.25) |
> | | | | | |
>
> - These results demonstrate that CCL achieves consistent improvements across different runs, indicating that its performance is stable and reproducible.
>
> > There is no ablation study on the latent representation, such as varying $dim(c)$ or analyzing the impact of reconstruction quality on retrieval and ICL performance, leaving hyperparameter sensitivity uncharacterized.
>
> - Thank you for highlighting this important point. We conducted an ablation study to investigate how sensitive CCL’s performance is to the dimension of the latent causal variable $c$. As shown in Table 2, even with a reduced latent dimension, CCL maintains strong performance across OOD benchmarks.
> - This suggests that CCL is not overly sensitive to the choice of $dim(c)$. Identifying an optimal dimension that ensures generalizable performance across domains — while minimizing memory footprint — is a promising direction for future research.
>
> Table 2. Accuracy on OOD benchmarks with varying $dim(c)$
> | dim( c )      | QNLI | PIQA  | WSC273 | YELP  | Avg. |
> | ----          | ---  | ----  | -----  | ----  | ---  |
> |  128          | 69.5 | **72.6**  | 56.1   | 93.5  | 72.9 |
> |  256          | **75.3** | 71.6  | 60.4   | 94.9  | 75.6 |
> |  1024 | 75.2 | 70.5  | **61.9**   | **95.4**  | **75.7** |
> | | | | | | |
>
> > All experiments focus on classification or short-answer tasks, so it remains unknown how CCL would extend to free-form generation or chain-of-thought prompting scenarios.
>
> - We appreciate this insightful comment. To explore the generalizability of CCL beyond classification, we conducted a new experiment on an unseen paraphrasing task, where the goal is to rewrite a negative sentence into one with an opposite (positive) sentiment.
> - While CCL was trained only on binary and multiple-choice classification tasks, it was applied here to a generation task. The system prompt instructed the model to “rephrase the sentence to express an opposite sentiment” — without explicitly directing it to convert negative sentences into positive ones.
>
> Table 3. Percentage of sentences that changed from negative to positive sentiment
> | phi4-mini-it    | Amazon | Yelp   |
> | ----            | -----  | -----  |
> | ICL (5-shot)     | 83.28  | 80.85  |
> | CCL (5-shot)     | **89.59**  | **84.77**  |
> | | | |
>
> - As shown in Table 3, CCL demonstrates superior performance compared to ICL, even on an unseen generation task. This suggests that CCL may hold promise for broader application beyond classification, potentially including text generation.
>
> We sincerely thank you for your thoughtful and constructive feedback.

---

> > ### Comment · Reviewer_Vb3H · 2025-08-06
> >
> > Thank you for the detailed clarifications in your rebuttal. I found your work to be valuable and appreciate the care you’ve taken in addressing the feedback.

---

> > > ### Author Response · Authors · 2025-08-08
> > >
> > > We sincerely appreciate your positive feedback once again and are glad our clarifications were helpful.

---

### Official Review · Reviewer_CojG · 2025-07-03

**Clarity:** 2
**Significance:** 3
**Originality:** 3
**Rating:** 5
**Confidence:** 4

**Summary:**

The paper introduces Causal-aware In-context Learning (CCL), a two-phase framework that first learns task-invariant latent causal variables with a VAE-style model and then selects demonstrations whose causal embeddings are closest to the query. Experiments on synthetic data, MGSM math problems, and four NLP benchmarks (QNLI, PIQA, WSC273, Yelp) across three LLMs verifie the effectiveness of proposed methods.

**Questions:**

NAN

**Ethical Concerns:**

["NO or VERY MINOR ethics concerns only"]

**Final Justification:**

Since I have not received a reply from the authors, I still have concerns regarding the generalization of the proposed framework, and only improved my score to 4.

**Limitations:**

NAN

**Paper Formatting Concerns:**

NAN

**Quality:**

3

**Strengths And Weaknesses:**

Strengths

Principled formula. Clear probabilistic model, disentangled latent factors, and derivation of an ELBO that removes the need to observe y at inference. These Principled formulation
 gives intuition and formal guarantees (within linear assumptions) for why causal retrieval helps.

Empirical evidence. This paper conduct on both synthetic and real datasets to very the effectiveness of proposed methods.

Weaknesses

Interpretability Challenge. While the paper adopts causal learning, the proposed latent causal variables lack interpretability. It remains unclear whether such simplified models can consistently learn domain-specific and causally invariant representations, particularly for complex tasks.


Insufficient Experiments. The paper lacks comparison with state-of-the-art learning-based and learning-free retrievers (e.g., [1–2]). Additionally, baseline details are vague, and the evaluation is limited to a narrow set of tasks. Experiments on generation tasks would further strengthen the work.

Writing and Clarity. The presentation would benefit from a clearer description of the complete algorithmic pipeline.

Need for Deeper Explanation. The method appears to leverage output labels (y) from the demonstration set in a way that resembles TTF [1]. A comparative analysis of the advantages and limitations of both approaches would be valuable.

The proposed method appears to have limited generalizability to tasks beyond those evaluated  (train on one task and test on the other task).


[1] Unraveling the Mechanics of Learning-Based Demonstration Selection for In-Context Learning. ACL 2025


[2] Compositional Exemplars for In-context Learning. ICML 2023.

---

> ### Author Rebuttal · Authors · 2025-07-31
>
> We sincerely appreciate your thoughtful and constructive review. Your comments reflect a deep understanding of our work, and we have made our best effort to address your concerns comprehensively. In doing so, we carefully referred to the papers you mentioned and conducted additional experiments to support our response.
>
> > Interpretability Challenge. While the paper adopts causal learning, the proposed latent causal variables lack interpretability.
>
> - We appreciate this insightful concern. In the vision domain, there has been significant research into causal representation learning using VAE-based models for OOD generalization [3,4].
> - However, to the best of our knowledge, **such approaches have not been widely studied in the natural language domain**. Our work aims to fill this gap by rigorously designing both the methodology and the objective function, and by demonstrating the validity of causal representation learning through controlled toy experiments.
> - That said, we agree with your suggestion that further analysis is needed to evaluate **whether a simple VAE structure can effectively disentangle domain-invariant and domain-variant features in natural language tasks**. In response, we conducted a qualitative analysis to interpret the meaning of the latent causal variables in our model.
> - As described in the paper, our goal is to decompose the latent features into components that are relevant to the task objective (domain-invariant) and components that are unrelated (domain-variant).
> - For instance, in a sentiment analysis task, domain-invariant features would correspond to expressions of sentiment (positive or negative), while domain-variant features might vary depending on the data source. For example, Yelp reviews may include aspects related to restaurant service or food quality, whereas Amazon reviews might focus on product quality or usability.
>
> - To qualitatively assess the interpretability of the latent variables $c$ (invariant) and $s$ (variant) in our VAE trained under the CCL framework, we visualized their effects via decoding. Specifically, given an input sentence embedding $x$, we inferred the corresponding $c$ and $s$. We then performed two decoding operations:
>   - Zeroing out $s$ to generate $x’_{s=0}$, and
>   - Zeroing out $c$ to generate $x'_{c=0}$.
>
> - Since $x_{s=0}'$ and  $x_{c=0}'$  lie in the same embedding space as $x$, we computed nearest neighbors based on embedding similarity to examine the semantics captured by $c$ and $s$.
>
> Qualitative Example 1 (Yelp dataset):
> Original negative sentence: “the red velvet pancakes were horrible and brown, and potatos were over cooked and bland.. would not recommend”
>
> Table 1. Top 5 nearest words
> | $x$                | $x'_{s=0}$         | $x'_{c=0}$ |
> | --------           | --------           | --------   |
> | horribleappetizers | unappetizing       | review     |
> | pancakes           | flavorless         | reviewers  |
> | potatos            | horribleappetizers | critiques  |
> | hadhorrible        | inedible           | soggy      |
> | bad                | trashed            | reviews    |
> |           |            |    |
>
> - In Table 1, $x$ retrieves words related to both sentiment and specific content (e.g., “pancakes”, “potatos”).  In contrast, $x_{s=0}'$ shows clusters of **strongly negative sentiment expressions**. Meanwhile, $x_{c=0}'$ surfaces terms such as “review” and “critiques,” suggesting that domain-variant features relate to the meta-characteristics of the dataset (e.g., user review context).
>
> Qualitative Example 2 (Amazon dataset):
> Original negative sentence: “Worked for about 4 months. DVD player will not eject or accept disks. Do not buy.”
>
> Table 2. Top 5 nearest words
> | $x$       | $x_{s=0}'$  | $x_{c=0}'$ |
> | --------  | --------    | -------- |
> | dvd       | unusable    | reverb   |
> | eject     | expired     | throw    |
> | disks     | cancelled   | film     |
> | unusable  | crappy      | review   |
> | purchased | trashed     | trip     |
> | |  |  |
>
> - Similarly, in Table 2, $x_{s=0}'$ captures **negative sentiment** towards product usability, while $x_{c=0}'$ retrieves terms more loosely associated with contextual metadata.
>
> - Our qualitative analysis indicates that CCL may learn to separate causal factors in natural language, even with a relatively simple VAE-based architecture.
>
> > Insufficient Experiments. The paper lacks comparison with state-of-the-art learning-based and learning-free retrievers (e.g., [1–2]).
>
> Thank you for pointing us to these valuable references. Both [1] and [2] offer thorough analyses of retrieval strategies under in-distribution ICL settings, and we will make sure to cite them appropriately in the revised manuscript.
>
> - We position CCL differently along two key dimensions:\
> - First, prior work typically assumes that the selection pool and the test query share the same distribution. While cross-task experiments are considered, they still assume distributional alignment between the test queries and the selection pool.
> - In contrast, CCL explicitly tackles the OOD setting, assuming a fixed demonstration pool across tasks and domains — which we consider to be a more practical and challenging scenario in many real-world situations. For example, datasets such as QNLI, PIQA, WSC273, and Yelp used in our evaluation do not appear in the demonstration pool.
> - Second, unlike prior approaches, CCL leverages domain-invariant features to guide retrieval. By explicitly modeling a causal factor $c$ that is intended to be invariant across domains, CCL aims to retrieve examples that are more likely to remain effective under OOD conditions.
>
> - We have also extended our evaluation to the ID setting. Table 3 reports the few-shot accuracy on SST-5 following the setup of [1]. CCL achieves competitive performance and outperforms other methods in the 5-shot and 10-shot settings.
>
> Table 3. Accuarcy performance comparision
> | SST-5 | 5-shot    | 10-shot   | 20-shot   |
> | ----  | -----     | -----     | -----     |
> | BERT  | 0.318     | 0.344     | 0.327     |
> | MLSM  | 0.322     | 0.349     | 0.332     |
> | EPR   | 0.383     | 0.390     | 0.369     |
> | TTF   | 0.385     | 0.390     | **0.427** |
> | CCL   | **0.397** | **0.422** | 0.416     |
> | | | |      |
>
> > Need for Deeper Explanation. The method appears to leverage output labels (y) from the demonstration set in a way that resembles TTF [1].
>
> - Thank you for this valuable observation. TTF uses a language model as a retriever often task-specific prediction modules for each task. TTF is based on the assumption that exemplars with similar labels to the predicted output of the test query are more useful for in-context learning.
>
> - In contrast, CCL uses labels only during training of the VAE — specifically, to help model the distribution of the output $y$ as a text embedding. This module is shared across tasks and does not require task-specific adaptation, allowing CCL to generalize to unseen tasks without additional retriever training.
>
> - While the latent representation used in TTF’s retriever output might superficially resemble our $c$ (since both are used to predict $y$), the underlying design philosophy is different. CCL’s $c$ is rooted in a causal framework: it is used not only for predicting $y$ but also for reconstructing $x$ as part of a principled data-generating process. This dual role is critical in ensuring that $c$ captures causal, invariant properties across domains.
>
> > The proposed method appears to have limited generalizability to tasks beyond those evaluated
>
> - We appreciate this comment. To evaluate generalization more rigorously, we conducted an additional experiment on an unseen generation task. Specifically, we tested the ability of CCL to support sentiment reversal paraphrasing, where the task is to transform a negative sentence into one with opposite (positive) sentiment.
>
> - CCL was trained only on binary and multiple-choice classification tasks. For the generation task, we used a system prompt that instructs the model to “rephrase the sentence to express an opposite sentiment” — without explicitly instructing it to convert negative sentences into positive ones.
>
> Table 4. Percentage of sentences that changed from negative to positive sentiment
> | phi4-mini-it    | Amazon | Yelp   |
> | ----            | -----  | -----  |
> | ICL (5-shot)     | 83.28  | 80.85  |
> | CCL (5-shot)     | **89.59**  | **84.77**  |
> | | | |
>
> - As shown in Table 4, CCL achieves better generalization on this unseen generation task, indicating its broader applicability beyond classification settings.
>
> > Writing and Clarity.
>
> - Thank you for your valuable suggestion to improve the clarity of the paper. We fully agree that including an algorithm will enhance readability, and we will incorporate it in the revised version accordingly.
>
> [3] Liu et al., Learning Causal Semantic Representation for Out-of-Distribution Prediction, NeurIPS 2021.\
> [4] Lu et al., Invariant Causal Representation Learning for Out-of-Distribution Generalization, ICLR 2021.

---

> > ### Comment · Reviewer_CojG · 2025-08-05
> >
> > Thank you for your response. The response has addressed most of my concerns.  I will improve my scores accordingly. For further communication, could you report the performance of your method and BERT on more difficult datasets, such as MMLU or coding genreation, or some other datasets to justify the generalization of your framework.  You can just sample some data for evaluation instead of evaluation on all data considering the ddl of response.

---

> ### Comment · Reviewer_CojG · 2025-08-09
>
> Since I have not received a reply from the authors, I still have concerns regarding the generalization of the proposed framework, and only improved my score to 4.

---

> > ### Author Response · Authors · 2025-08-09
> >
> > We are truly delighted that our response has resolved the concerns you had. Moreover, the points you raised gave us the opportunity to reflect and discuss further, which enriched our research.
> >
> > Before proceeding with our detailed response, we would like to apologize for the slight delay due to the ongoing experiments, and we kindly ask for your understanding.
> >
> > We are also very pleased to address your additional questions. **We applied CCL to the MMLU task as well, and also considered using BERT as the language model for inference.**
> >
> > - We adopted MMLU\'s common practice of using a fixed 5-shot setting for the few-shot size. Specifically, we used the "lukaemon/mmlu" dataset available on Hugging Face. This dataset consists of QA pairs across 57 diverse domains, with samples from each domain\'s train set and validation set available for use as few-shot examples. The train set contains the samples used for the fixed 5-shot setting, while the validation set contains around 30 samples per domain on average (though the exact number varies by domain).
> >
> > - We combined the train set and validation set to train CCL. In other words, in the MMLU task setting with 57 environments, we trained a VAE. Afterward, for each input query, we collected 5 examples at the $c$ embedding level regardless of the domain. This approach is based on the fact that CCL is inherently designed to identify the intent of a query and retrieve examples that are helpful for task performance, irrespective of the domain. (We also retrieved 5 examples using the $x$ embedding, and we refer to the results from this approach as ICL.)
> >
> > Table 5. MMLU performance comparison
> >
> > | Method | Avg. Acc |
> > | --- | --- |
> > | ZS | 60.48 |
> > | fewshot | 61.37 |
> > | ICL | 61.37 |
> > | CCL | **61.52** |
> >
> > - Table 5 shows the accuracy on MMLU when using Phi4-mini-IT as the base model. We observe that CCL shows a slight performance improvement compared to other baselines.
> >
> > - We also conducted performance evaluation on BERT in Table 6.
> >
> > Table 6. MMLU performance comparison based on BERT
> >
> > | Method | Avg. Acc |
> > | --- | ---|
> > | ZS | 23.11 |
> > | fewshot | 23.17 |
> > | ICL | 23.14 |
> > | CCL | **23.31** |
> >
> > - In BERT, the effect of few-shot in-context learning does not appear to be as pronounced as in other decoder-based models (e.g. Phi4-mini-IT), **but we could still observe that CCL yields a slight performance improvement.** One possible reason for the limited effect of few-shot in-context learning is that BERT has a maximum sequence length of 512, which may have caused the 5 examples to be truncated and not fully included in the input.
> >
> > - **In summary, as you suggested, we conducted experiments using the more complex MMLU benchmark and the transformer encoder based BERT model.** The MMLU experiments aimed to test CCL on a more challenging task, while using BERT as the base model was intended to examine whether the examples selected by CCL could perform well regardless of the underlying model architecture.
> >
> > - **A common finding across the two experiments is that CCL consistently demonstrated slight but stable performance improvements over the baselines. These results support the robustness and domain-agnostic applicability of CCL in enhancing few-shot in-context learning.**
> >
> > - We note that these experiments were carried out within the limited time of the discussion period, so further experiments and more detailed analyses are still needed. **We will address these points in more detail in the revised version of the manuscript.**
> >
> > We truly appreciate your thoughtful and helpful review.

---

### Author Response · Authors · 2025-08-04

## Review Summary
We sincerely thank all reviewers for their thoughtful and detailed feedback. Your comments have helped us sharpen both the theoretical foundations and the empirical evaluation of our work. Below, we briefly summarize the strengths and weaknesses pointed out by reviewers.

1. Strengths
- Causal-aware In-Context Learning: Reducing Spurious Correlations and Ensuring Robustness to Distribution Shifts (Reviewer CojG, 2GL1, Vb3H, 1SuJ)
- Rigorous Theoretical Foundations & Guarantees (Reviewer CojG, Vb3H, 1SuJ)
- Consistent Empirical Gains Across Synthetic & Real OOD Tasks (Reviewer CojG, Vb3H, 1SuJ)

2. Weakness
- Unclear Interpretability and Identifiability of Causal Factors (Reviewer CojG, 2GL1, 1SuJ)
- Broader & Statistically-Rigorous Evaluation (Reviewer CojG, Vb3H, 1SuJ)
- Need for More Detailed & Clear Methodology Description (Reviewer CojG, 2GL1, 1SuJ)

## General Response
Our research, CCL, retrieves demonstrations by **causal** rather than surface-level similarity, cutting spurious correlations and **staying robust when the distribution shifts**. This causal framing has formal guarantees and, in both synthetic and real OOD tests, consistently outperforms standard retrievers—showing that focusing on underlying causal factors is both principled and effective.

- To demonstrate that **CCL learns meaningful causal features**, we included a qualitative decoding analysis:
Procedure: Given an input embedding $x$, we infer $(c,s)$ with our VAE, then decode two counterfactual embeddings—$x_{s=0}'$ (keep $c$, remove $s$) and $x_{c=0}'$ (keep $s$, remove $c$)—and inspect their nearest neighbors.
Finding: Across both Yelp and Amazon reviews, $x_{s=0}'$ consistently retrieves sentiment-laden words (“unappetizing”, “expired”), whereas $x_{c=0}'$ surfaces dataset-specific meta-tokens (“review”, “critique”). This confirms that **$c$ captures domain-invariant meaning**, while **$s$ captures domain-variant meaning**. (Reviewer CojG, 2GL1, 1SuJ)

- We conducted a wide range of experiments and **added repeated trials** to assess the model’s stability. Specifically, we reran the same OOD benchmarks with multiple random seeds and confirmed consistent performance gains; we also **varied the dimensionality of the latent causal variable $c$** and found that results changed little, demonstrating **robustness to hyper-parameter choices** (Reviewer Vb3H). In addition, we conducted an **unseen generation task**—flipping sentiment from negative to positive—to demonstrate CCL’s competitiveness on generation tasks (Reviewer CojG, Vb3H), and we verified that it **remains competitive** with the latest retrievers, even in **in-distribution settings** such as SST-5, confirming its broad competitiveness (Reviewer CojG).

- To meet the request for a more detailed and clearer methodology, we took a concrete steps in the revision. We **timed every stage of the pipeline**—VAE training, latent-$c$ extraction, demonstration retrieval, and LLM decoding—and confirmed that the **overall latency remains comparable to ICL**, showing CCL’s practical overhead is negligible (Reviewer 2GL1).

Once again, we thank the reviewers for their thoughtful comments and the opportunity to strengthen this work. We hope our responses address the reviewers’ concerns, and we remain happy to clarify any further questions or share additional insights into our research.

---

### Note · Authors · 2025-08-13

We would like to express our sincere gratitude to the AC for their dedicated efforts in handling the review process, and to the reviewers for their valuable commitment.

We appreciate that the reviewers recognized the strengths and novelty of our work:
- The necessity of CCL: CCL addresses the bias of $x$-embedding–based selection toward surface-level similarity in OOD. Grounded in a causal framework, it infers domain-invariant $c$ embeddings, improving OOD performance.
- Theoretical and experimental validation: Thm 3.3 \& 3.4 shows that $c$-based selection can achieve lower test error than $x$-based selection under distribution shift. Synthetic and diverse NLP task results further confirm CCL’s effectiveness in OOD setting.


We also summarize the our responses to the concerns raised by the reviewers in rebuttal:
- Interpretation of causal features: We performed a qualitative analysis during the rebuttal period, confirming that $c$ capture domain-invariant meaning, while $s$ captures domain-variant meaning.
- Additional experiments: We conducted all reviewer-recommended experiments during rebuttal.
  - Comparison with recent baselines: We verified performance against the reviewer-suggested baseline and clarified methodological differences.
  - Unseen generation task: Using the same CCL as in Table 3, we tested paraphrasing sentences into the opposite sentiment. CCL outperformed standard ICL in this unseen generation setting.
  - Repeated-trial and latent dimension robustness: Across multiple seeds and latent dimensions, CCL showed consistent reproducibility, stability, and robustness.
  - Training and inference time comparison: Once trained, CCL can pre-extract $c$, adding no inference cost. Measurements show training/inference time and parameter overhead are reasonable.

In the discussion phase:
- Applicability to more challenging tasks and to BERT: Our experiments showed that CCL outperformed standard ICL on MMLU. When applied to BERT, CCL also showed modest performance improvements.
- On the simplicity of VAE architecture: We conducted a multi-hop QA task, demonstrating that CCL remains effective even for complex reasoning tasks.

During the rebuttal, the concerns of Reviewer Vb3H and 1SuJ were resolved, while Reviewer CojG and 2GL1 raised additional questions in discussion. We conducted further experiments that addressed and resolved their concerns.

We once again thank the ACs for their efforts and the reviewers for their constructive feedback.

---

### Decision · Program_Chairs · 2025-09-17

**Decision:**

Accept (poster)

**Comment:**

This paper introduces Causal-aware In-context Learning (CCL), which is a principled framework that integrates causal representation learning into in-context learning to improve out-of-distribution generalization. The work is theoretically well-grounded (Theorems 3.3 and 3.4 show the benefits of causal retrieval over surface similarity) and supported by experiments across synthetic, multilingual math reasoning, and diverse NLP benchmarks. The reviewers initially raised concerns regarding interpretability of causal factors, baseline coverage, and robustness. During rebuttal, the authors provided useful information including qualitative analyses and tests on more challenging benchmarks such as MMLU and generation tasks. I feel that these clarifications and new results resolved most issues. I would recommend acceptance.